∂ | **Open Peer Review** | Clinical Microbiology | Research Article

# Salivary microbiota reflecting changes in subgingival microbiota

Jae-Suk Jung,[1] Joong-Ki Kook,[2] Soon-Nang Park,[2] Yun Kyong Lim,[2] Geum Hee Choi,[1] Sunjin Kim,[1] Suk Ji[1]

**ABSTRACT** The purpose of this study was to determine whether subgingival microbial changes according to the severity of periodontal disease and following the non-surgical periodontal treatment of periodontitis are reflected in the saliva microbiota. Subgingival and saliva samples were collected from 7 periodontally healthy controls, 14 patients with gingivitis, 12 with moderate periodontitis, and 18 with severe periodontitis. Among subjects who received treatment, seven moderate and seven severe periodontitis patients were selected for post-treatment microbial analysis, and their samples were analyzed at baseline and 6 months after treatment. The V3 and V4 regions of the 16S rRNA gene were sequenced, and correlations of the relative abundance of phyla and health- or periodontitis-dominant species between subgingival plaque and saliva samples were analyzed using Spearman signed-rank tests. Alpha diversity was higher in saliva than subgingival plaque samples, and beta diversity was significantly different between the two samples. However, levels of phyla and most health- or periodontitis-dominant species in salivary microbiota were significantly correlated with those in subgingival plaque. The improvement in clinical parameters following treatment was accompanied by a microbial shift not only in subgingival plaque but also in saliva. The abundance of 2 phyla including Bacteroidetes, 6 genera including *Porphyromonas* and *Treponema*, and 11 species including *Porphyromonas gingivalis*, *Tannerella forsythia*, and *Filifactor alocis* was significantly reduced in saliva following treatment. These results indicate that the salivary microbiota can reflect changes in the subgingival microbiota, suggesting that saliva can be used as a diagnostic tool to monitor the periodontal health status of individuals.

**IMPORTANCE** The salivary microbiota has attracted increasing attention as a promising method for monitoring periodontal disease. With regard to the pathogenesis of periodontal disease, however, subgingival plaque microbiota is the dominant etiological factor. Although it has been established that periodontopathogenic bacteria exist in saliva and their distribution differs, depending on the severity of the disease, it is necessary to analyze the extent to which the salivary microbiota reflects the subgingival microbiota. This study explored whether subgingival microbial changes according to the severity of periodontal disease and following the non-surgical periodontal treatment of periodontitis are reflected in the saliva microbiota and concluded that the salivary microbiota can reflect changes in the subgingival microbiota. Saliva can be used as a diagnostic tool to monitor the periodontal health status of individuals.

**KEYWORDS** periodontal disease, saliva, subgingival plaque, microbiota, *Porphyromonas gingivalis*, *Tannerella forsythia*, *Filifactor alocis*

Periodontal disease is an inflammatory disease that affects the supporting structures of the teeth and is characterized by the host's immune response to microorganisms in the subgingival space (1–3). The disease typically starts with gingivitis, characterized by inflammation localized to the gingiva, which, if left untreated, can progress to periodontitis, defined by resorption of alveolar bone.

Address correspondence to Suk Ji, sukji@ajou.ac.kr.

Jae-Suk Jung and Joong-Ki Kook contributed equally to this article. Author order was determined based on alphabetical order.

The authors declare no conflict of interest.

See the funding table on p. 14.

The symbiotic relationship between the oral microbiota and host immune response is crucial for maintaining periodontal health. Disruptions in this balance, often due to plaque deposits on the tooth surface, can lead to an overgrowth of pathogenic bacteria and inflammation of periodontal tissues (2). The host inflammatory reaction plays a pivotal role in dysbiosis of the subgingival microbiota, and polymicrobial biofilm dysbiosis and periodontal inflammation drive chronic periodontitis. This dysbiosis involves an imbalance in the types and proportions of bacterial species present, often leading to an increase in pathogenic or disease-associated bacteria, such as red complex (4). Periodontal treatment, including mechanical removal of subgingival plaque on the tooth surface, can manage periodontal inflammation and evoke changes in the subgingival microenvironment related to changes in the composition and proportions of the bacterial species in the polymicrobial biofilm that characterize dysbiosis (5).

The salivary microbiota has attracted increasing attention as a promising method for monitoring periodontal disease because saliva can be collected easily, repetitively, and non-invasively (6–8). With regard to the pathogenesis of periodontal disease, however, subgingival plaque microbiota, not saliva, is the dominant etiological factor. Subgingival microbiota is only one of several sources of salivary microorganisms. Saliva secreted sterilely from the salivary glands can contain diverse bacterial species shed from various oral sites with different ecological conditions, such as the tongue dorsum, buccal mucosa, and tooth surface, on its way to the oral cavity (9). Several studies have shown that the bacterial composition of saliva is significantly different from that of the teeth, including both supra- and subgingival plaque microbiota, and is closer to that on mucosal surfaces such as the tongue coating (10–12). Yamanaka et al. showed that the community structure of salivary microbiota is distinct from that of supragingival plaque and that the salivary microbiota is compositionally stable against supragingival microbiota shifts (13). Nevertheless, it is well known that saliva contains various periodontopathic bacteria (14–17). Our recent study showed that as the severity of periodontal disease increased, levels of periodontopathic bacterial species increased, while those of health-associated bacterial species decreased in saliva (14). Although it has been established that periodontopathogenic bacteria exist in saliva and their distribution differs, depending on the severity of the disease, it is necessary to analyze the extent to which the salivary microbiota reflects the subgingival microbiota. Monitoring the relationship between the subgingival and salivary microbiota may provide insights into the oral health status of individuals and aid in the early detection of periodontal disease.

The purpose of this study was therefore to determine whether saliva microbial changes reflect changes in the subgingival plaque microbiota. For this purpose, subgingival plaque and salivary microbiota of subjects were simultaneously analyzed according to the severity of periodontal disease and before and after non-surgical periodontal treatment for periodontitis.

## MATERIALS AND METHODS

### Subjects and non-surgical periodontal treatment

These participants correspond to a subgroup analyzed in our previous study (14). Twenty-one of 72 subjects in the previous study were excluded in this study because not all subgingival plaque samples underwent 16S rDNA amplicon sequencing. Subject information is described in detail in our previous study (14). Briefly, participants had no history of systemic disease that could influence the prognosis of periodontitis, untreated caries, or orthodontic appliances. None of the subjects were pregnant/breastfeeding, heavy smokers (>10 cigarettes per day), or had taken antibiotics, antimicrobials, and/or anti-inflammatory drugs during the 3 months prior to examination and sampling. All participants underwent a full-mouth periodontal examination followed by saliva and subgingival plaque sampling by a single trained periodontist. Participants were classified into four groups: periodontally healthy control [health (H) group], gingivitis (G) (G group),

moderate periodontitis (MP) (MP group), and severe periodontitis (SP) (SP group) groups based on their periodontal status in accordance with the clinical criteria stated in the consensus report of the World Workshop in Periodontics (18).

For moderate and severe periodontitis patients, non-surgical periodontal treatment was performed. For moderate periodontitis subjects, scaling and one or two deep cleanings without local anesthesia were performed, while for severe periodontitis patients, one scaling and four non-surgical root debridements under local anesthesia were performed. Measurement of clinical parameters and sampling of saliva and subgingival plaque were conducted 1, 3, and 6 months after treatment. We analyzed the degree of improvement in clinical parameters after 1, 3, and 6 months and preferentially selected subjects for post-treatment sequencing based on the degree of improvement in mean probing depth (PD) and number of sites with a PD of 5 mm. Subjects with ≤4 sites with a PD of ≥5 mm were selected for the reason that the clinical endpoint of "≤4 sites with PD of ≥5 mm" is considered to be effective in determining disease remission/control after active periodontal treatment (19). All were selected from among the non-smoking patients. Then, 16S rDNA amplicon sequencing was performed on pre-treatment and on post-treatment time points to show the significant improvements in clinical parameters.

## Saliva and subgingival plaque sampling

Saliva was collected as described previously using the mouth rinse method (16). Briefly, saliva samples were collected after gargling for 1 min with 10-mL normal saline solution. All participants were instructed to not ingest anything, rinse their mouths, or take any oral hygiene measures for 1 h prior to sampling. Subgingival samples from the H group were collected for teeth with a PD of ≤3 mm and bleeding on probing (BOP) (+) in at most two out of six sites, and those from the G group were collected for teeth showing a PD of ≤3 mm and BOP (+) in at least three of six sites. Subgingival samples from the MP and SP groups were collected from teeth with a PD of ≥4 mm and BOP (+) for at least two of the six sites. Subgingival plaque samples were obtained from four sites of each tooth (mesiobuccal, mesiolingual, distobuccal, and distolingual sites) using absorbent paper strips (Oraflow, Smithtown, NY, USA). Supragingival plaque on the tooth surface was carefully removed with curettes, and each tooth site was gently dried for 10 s with compressed air. A paper strip was inserted carefully into the gingival sulcus/pocket until mild resistance was felt and left in place for 30 s. The strip was then transferred into a microtube containing 200 µL of phosphate-buffered saline. Samples were centrifuged at $15,928 \times g$ for 5 min, and pellets were stored at $-80°C$ until DNA extraction.

## DNA extraction, amplification of the 16S rRNA gene, and Illumina sequencing

DNA extraction and sequencing were performed at ChunLab, Inc (https://www.cjbio-science.com/) as described in our previous study (14). Briefly, DNA was extracted using a FastDNA SPIN Kit for soil (MPBIO), and PCR amplification was performed using primers targeting the V3 and V4 regions of 16S rDNA. PCR products were sequenced with an Illumina MiSeq sequencing system at ChunLab, Inc. UCHIME (20). Raw sequence data (FASTQ or FASTA format) were uploaded to www.ezbiocloud.net; taxonomic profiles were generated from sequencing data; and profiles from different samples were grouped and compared. Data for this study have been deposited in the European Nucleotide Archive at European Molecular Biology Laboratory–European Bioinformatics Institute under accession number PRJEB61123.

## Statistical analysis

Means of plaque index (PI), PD, clinical attachment level, modified sulcus bleeding index, gingival index, and the BOP (%) of the full mouth and teeth with subgingival plaque were compared among the four groups using the Kruskal–Wallis test. A linear mixed-effects model was used to estimate the effect of time on changes in each clinical parameter following treatment. Microbiome taxonomic profiling and comparisons among the four

groups and between pre- and post-treatment were performed using BIOiPLUG (https://www.ezbiocloud.net), a web-based life information analysis cloud platform provided by ChunLab, Inc. The analysis was based on the relative abundance of each taxonomic group. Alpha diversity based on ACE, Chao 1, Jackknife, and the number of species identified in subgingival plaque vs saliva samples or pre- vs post-treatment was analyzed using the Wilcoxon rank-sum test. Beta diversity distances based on Jensen–Shannon (21) were analyzed. Kruskal–Wallis *H* statistical analysis was conducted to evaluate the significance of differences in the dominant taxa among the four groups and between pre- and post-treatment periodontitis groups. Correlations between the relative abundances of phyla and bacterial species in subgingival plaque and saliva samples were analyzed using Spearman signed-rank tests. All statistical analyses were performed using SAS (version 9.4; SAS Institute Inc., Cary, NC, USA) and the R package (version 4.1.2; R Foundation for Statistical Computing, Vienna, Austria). Results were considered statistically significant when *P* values were less than 0.05.

## RESULTS

### Sample groups and clinical characteristics

A total of 51 subjects—7 subjects in the H group, 14 patients in the G group, 12 patients in the MP group, and 18 patients in the SP group—were enrolled in this study. There were no significant differences in mean age ($P = 0.041$) among groups, whereas gender distribution ($P = 0.001$) and smoking status ($P = 0.001$) showed significant differences among the four groups. Nine light smokers were included in the SP group only. Clinical parameters of the four groups were differentiated by teeth in the entire mouth and the tooth from which subgingival plaque was harvested and are detailed in Table 1. There were statistically significant differences in all clinical parameters except the number of teeth among the four groups (Table 1).

### Taxa diversity between subgingival plaque and saliva samples

When alpha diversity metrics were applied across 51 subjects using ACE, Chao 1, Jackknife, in addition to the number of identified species, diversity was found to be significantly higher in saliva samples (Fig. 1A). Alpha diversity for the H, G, MP, and SP groups tended to be higher in saliva than subgingival plaque samples (Supplement 1). Beta diversity of clustering analysis based on Jensen–Shannon divergence at the species

**TABLE 1**  Clinical characteristics of the entire mouth and tooth from which subgingival plaque was sampled[a,b]

| | | Health (n = 7) | Gingivitis (n = 14) | Moderate periodontitis (n = 12) | Severe periodontitis (n = 18) | P value |
|---|---|---|---|---|---|---|
| General characteristics | Age (years) | 49.14 ± 6.49 | 42.9 ± 3.1 | 43.5 ± 3.8 | 48.8 ± 2.6 | 0.041 |
| | Male/female | 1/6 | 0/14 | 7/5 | 12/6 | <0.001 |
| | Non-smoker/current smoker | 7/0 | 14/0 | 12/0 | 9/9 | <0.001 |
| Full mouth | PI | 0.21 ± 0.04 | 0.6 ± 0.1 | 1.2 ± 0.1 | 1.1 ± 0.1 | <0.001 |
| | PD | 1.90 ± 0.05 | 2.3 ± 0.1 | 2.7 ± 0.0 | 3.7 ± 0.1 | <0.001 |
| | CAL | 2.24 ± 0.11 | 2.4 ± 0.1 | 2.8 ± 0.0 | 4.2 ± 0.2 | <0.001 |
| | mSBI | 0.18 ± 0.02 | 0.7 ± 0.1 | 1.0 ± 0.1 | 1.2 ± 0.1 | <0.001 |
| | GI | 1.46 ± 0.03 | 1.8 ± 0.1 | 2.1 ± 0.1 | 2.2 ± 0.1 | <0.001 |
| | BOP (%) | 18.10 ± 1.57 | 51.2 ± 5.7 | 67.2 ± 4.1 | 74.0 ± 3.9 | <0.001 |
| | Number of teeth | 27.29 ± 0.29 | 26.6 ± 0.5 | 27.7 ± 0.2 | 26.2 ± 0.5 | 0.082 |
| Tooth–subgingival plaque | PI | 0.00 ± 0.00 | 0.7 ± 0.2 | 1.5 ± 0.2 | 1.2 ± 0.1 | <0.001 |
| | PD | 2.10 ± 0.08 | 2.5 ± 0.1 | 3.2 ± 0.1 | 4.5 ± 0.2 | <0.001 |
| | CAL | 2.38 ± 0.22 | 2.7 ± 0.1 | 3.3 ± 0.1 | 5.5 ± 0.3 | <0.001 |
| | mSBI | 0.20 ± 0.06 | 1.0 ± 0.1 | 1.6 ± 0.2 | 1.5 ± 0.1 | <0.001 |
| | GI | 1.57 ± 0.17 | 2.0 ± 0.1 | 2.3 ± 0.1 | 2.4 ± 0.1 | <0.001 |

[a]Values are presented as mean ± standard deviation. The *P* values were obtained by the Mann–Whitney *U* test. A *P* value of <0.05 was considered to indicate statistical significance.
[b]BOP, bleeding on probing; CAL, clinical attachment level; GI, gingival index; mSBI, modified sulcus bleeding index; PD, probing depth; PI, plaque index.

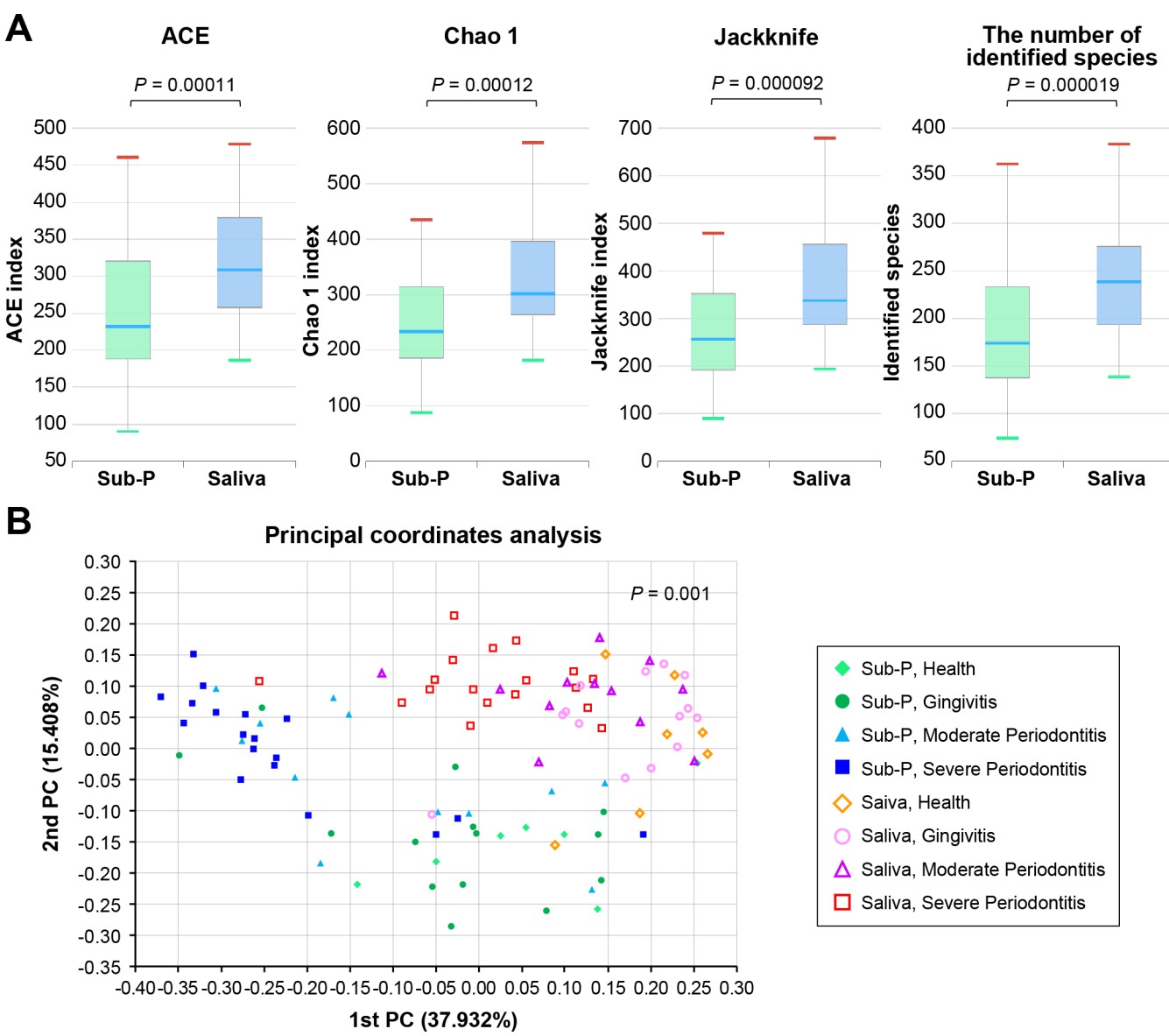

**FIG 1** Comparison between subgingival plaque and salivary microbiota. (A) Comparison of ACE, Chao1, Jackknife, and the number of identified species. Subgingival plaque and saliva samples were collected from a total of 51 subjects with varying severities of periodontal disease. Each value is presented as a box plot. Top, middle, and bottom lines of the boxes represent the 25th, 50th (median), and 75th percentiles, respectively. The significance of differences between two groups was evaluated using the Wilcoxon rank-sum test, and *P* < 0.05 was considered to indicate statistical significance. (B) Principal coordinate analysis (PCoA) plot illustrating beta diversity distance matrices of the Jensen–Shannon distance comparing the sample distribution between the two groups. The filled dots represent subgingival plaque samples, and the empty dots represent saliva samples. Permutational multivariate analysis of variance results demonstrated beta set significance (*P* = 0.001) between the subgingival plaque and saliva groups. Sub-P, subgingival plaque.

level, analyzed across 51 subjects and by disease severity, showed different communities in the subgingival plaque and saliva samples [permutational multivariate analysis of variance (PERMANOVA) results, *P* = 0.001] (Fig. 1B). Clustering analysis by severity of periodontal disease showed that salivary and subgingival communities tended to cluster separately as severity increased. In healthy individuals, the subgingival and salivary microbial communities were mixed (*P* = 0.043), but a clear tendency toward separation emerged as severity worsened (*P* < 0.001 of the SP group, Supplement 2).

## Microbial correlations between subgingival plaque and saliva samples according to the severity of periodontal disease

The distribution of the six major phyla in subgingival plaque and saliva samples was analyzed, as was the correlation of phyla between these two sample types. As the severity of periodontal disease increased, Firmicutes, Proteobacteria, and Actinobacteria tended to decrease, while Bacteroidetes, Fusobacteria, Spirochaetes, and Synergistetes tended to increase in both subgingival plaque and saliva samples (Fig. 2A). These shifts were consistent with what we observed in our previous study (14). Fluctuations of these six phyla according to the severity of periodontal disease, with the exception of Fusobacteria, were significantly correlated with each other (Fig. 2A). A total of approximately 1,000 species were analyzed in subgingival plaque and saliva samples, and 17–22 species with an abundance of more than 1% in any of the subgingival plaque and saliva samples accounted for more than 56%–77% of all bacterial species. Among them, only three species in the *Haemophilus parainfluenzae* group, *Fusobacterium nucleatum* group, and *Streptococcus pneumoniae* group were found in more than 1% of all four groups in subgingival plaque and saliva samples (Fig. 2B and C). Examination of the bacterial species present at greater than 1% in any of the four groups revealed 13 species increased (periodontitis dominant), whereas 7 species decreased (health dominant) as the severity of periodontal disease increased (Supplement 3). When examining the correlation for these species between subgingival plaque and saliva samples, most species except those in the *F. nucleatum* group showed a positive correlation. These included *Porphyromonas gingivalis* ($r = 0.811$), *Tannerella forsythia* ($r = 0.635$), *Filifactor alocis* ($r = 0.814$), *Rothia dentocariosa* ($r = 0.711$), and *Lautropia mirabilis* ($r = 0.619$) (Table 2; Supplement 4). These results indicate that the distribution of periodontitis- or health-dominant species in subgingival plaque is correlated with that of saliva.

## Comparison of clinical characteristics before and after treatment of subjects with moderate to severe periodontitis

Seven patients with moderate periodontitis with a mean age of 46.29 years and seven patients with severe periodontitis with a mean age of 51.71 years were selected for microbial analysis following treatment. Clinical parameters pre-treatment and 1, 3, and 6 months after treatment are detailed in Table 3. Gradual clinical improvement in all parameters was observed up to 6 months following non-surgical treatment (Table 3).

## Microbial characteristics of subgingival and saliva samples before and after periodontitis treatment

To determine whether changes in microbial distribution were observed not only in subgingival plaque, which is the target site of non-surgical periodontal treatment, but also in saliva, microbial shifts pre- and post-treatment were compared for each of the subgingival plaque and saliva samples. Because the extent of clinical improvement was most evident after 6 months of treatment (Table 3), subgingival plaque (Fig. 3) and saliva (Fig. 4) samples at that time point were analyzed using 16S rDNA amplicon sequencing. The comparison pre- and post-treatment was performed for a total of 14 subjects with periodontitis to clarify and simplify the results.

Alpha diversity was significantly higher pre-treatment compared with post-treatment in both subgingival plaque and saliva samples (Fig. 3A and 4A). Clustering analysis at the species level showed a significant difference between pre- and post-treatment saliva samples (Fig. 4B; PERMANOVA results, $P = 0.02$). Relative abundances of the taxa that differed significantly pre- and post-treatment for all 14 subjects are presented in Fig. 3C through E and 4C through E. Additionally, to assess the extent of changes after treatment in the moderate and severe periodontitis groups, the relative abundances of the taxa are displayed along with those in the H and G groups (Supplements 5 and 6). Levels of Fusobacteria and Synergistetes in subgingival plaque (Fig. 3C) samples and Bacteroidetes and Spirochaetes in saliva (Fig. 4C) samples were higher pre-treatment than

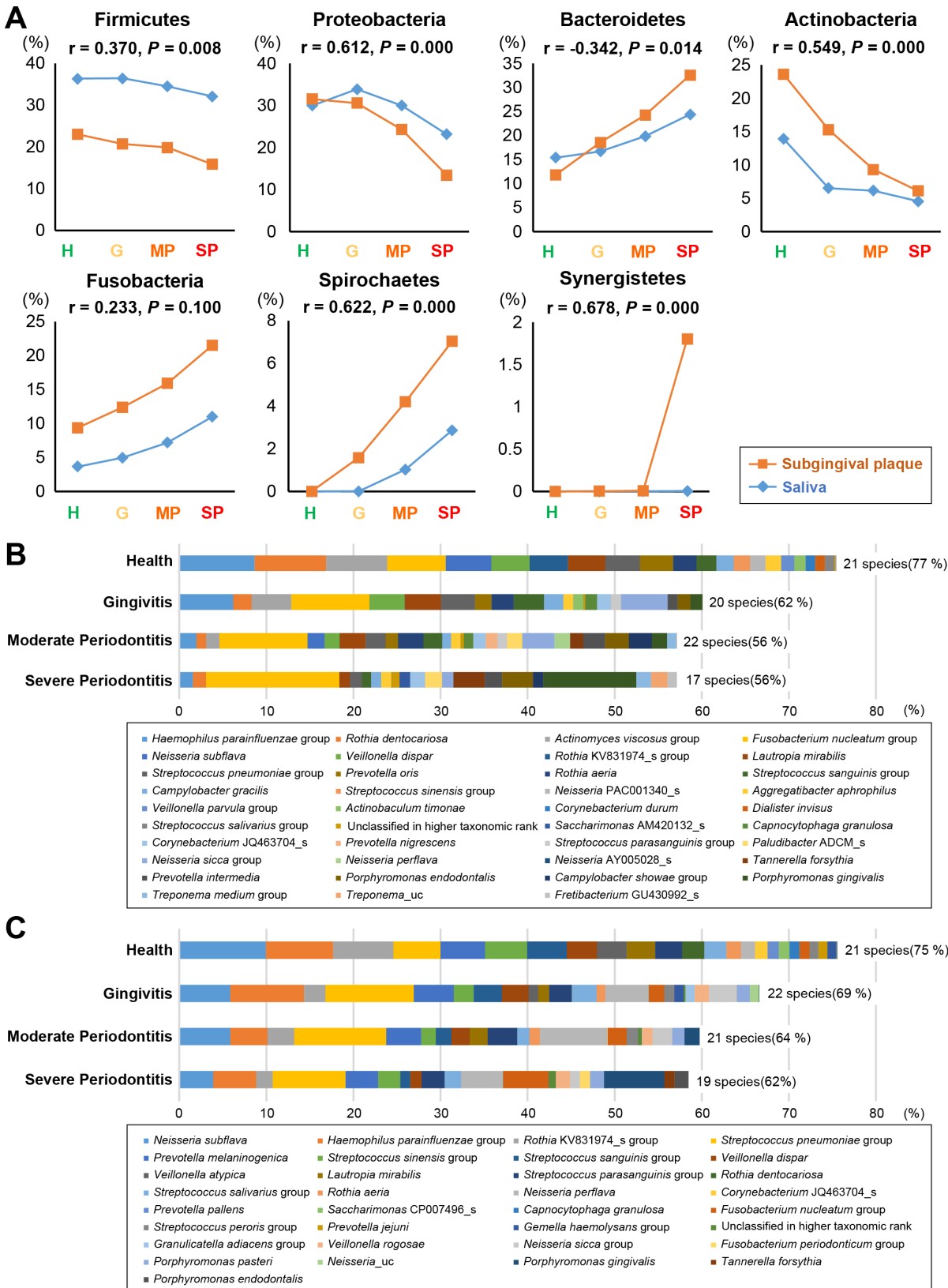

**FIG 2** Taxa distribution of subgingival plaque and saliva at the phylum (A) and species (B) levels according to the severity of periodontal disease. (A) Correlation between the relative abundance of seven phyla in subgingival plaque and saliva samples. Correlation analysis was performed by comparing the relative abundance of the seven phyla in subgingival plaque samples to those in saliva samples from a total of 51 subjects comprising 7 periodontally healthy subjects (Continued on next page)

Fig 2 (Continued)

(H), 14 patients with G, 12 patients with MP, and 18 patients with SP. The analysis was generated using the "Spearman" method and the Spearman correlation coefficient, *r*, and *P* values are indicated at the top of the graph. *P* < 0.05 was considered to indicate statistical significance. (B and C) Relative abundance of species comprising more than 1% of subgingival plaque (B) and saliva (C) samples. Descriptions to the right of the bars are the numbers and distribution (%) of species comprising more than 1% in each group according to the severity of periodontal disease. G, gingivitis; H, health; MP, moderate periodontitis; SP, severe periodontitis.

post-treatment. These phyla, in terms of distribution by the severity of disease, all showed a tendency to increase as the disease severity increased (Fig. 2A). Levels of six genera in subgingival plaque, including *Fusobacterium* and *Porphyromonas*, significantly decreased following treatment, while levels of six genera including *Treponema* and *Porphyromonas* significantly decreased in saliva samples following treatment (Fig. 3D and 4D, respectively). Specifically, the reduction in *Porphyromonas* in saliva after treatment correlated with that in subgingival plaque (Supplement 7). In comparison to H or G groups, the levels of reduction of periodontitis-dominant taxa seemed to be more remarkable in subgingival plaque than in saliva; in subgingival plaque, periodontitis-dominant taxa were observed at lower levels than in the G group, while in saliva, levels of these bacteria were similar or higher than those in the G group, especially after treatment of severe periodontitis (Supplements 5 and 6). Levels of *P. gingivalis*, *T. forsythia*, and *F. alocis* in both subgingival plaque and saliva samples significantly decreased following treatment (Fig. 3E and 4E). Treatment reduced the presence of these three bacterial species in each of the 14 subjects to less than 0.05%, with the exception of a few subjects (Fig. 5). Levels of reduction of the three species by treatment were more remarkable in subgingival plaque than in saliva; levels of *P. gingivalis*, *T. forsythia*, and *F. alocis* were reduced by 113-fold (3.87% vs 0.03%), 5-fold (1.53% vs 0.31%), and 15-fold (0.35% vs 0.02%) in subgingival plaque, while in saliva, levels decreased by 7.3-fold (5.39% vs 0.74%), 3.6-fold (1.10% vs 0.31%), and 5.0-fold (0.31% vs 0.06%), respectively. The extent of reduction of these three bacterial species in saliva did not show a positive correlation

**TABLE 2**  Correlation of relative abundance of health- or periodontitis-dominant bacterial species between subgingival plaque and saliva

|  | Taxon name | Spearman correlation coefficient (*r*) | *P* value |
| --- | --- | --- | --- |
| Periodontitis-dominant bacterial species | *Porphyromonas gingivalis* | 0.811 | 0.000 |
|  | *Fusobacterium nucleatum* group | 0.198 | 0.164 |
|  | *Tannerella forsythia* | 0.635 | 0.000 |
|  | *Porphyromonas endodontalis* | 0.661 | 0.000 |
|  | KE332528_s | 0.844 | 0.000 |
|  | *Prevotella intermedia* | 0.702 | 0.000 |
|  | Treponema_uc | 0.801 | 0.000 |
|  | ADCM_s | 0.560 | 0.000 |
|  | *Treponema medium* group | 0.520 | 0.000 |
|  | *Campylobacter showae* group | 0.402 | 0.003 |
|  | GU430992_s | 0.668 | 0.000 |
|  | *Mycoplasma faucium* | 0.800 | 0.000 |
|  | *Filifactor alocis* | 0.814 | 0.000 |
| Health-dominant bacterial species | *Haemophilus parainfluenzae* group | 0.544 | 0.000 |
|  | *Rothia dentocariosa* | 0.711 | 0.000 |
|  | *Lautropia mirabilis* | 0.619 | 0.000 |
|  | *Neisseria subflava* | 0.346 | 0.013 |
|  | KV831974_s group | 0.564 | 0.000 |
|  | *Streptococcus sanguinis* group | 0.518 | 0.000 |
|  | *Streptococcus sinensis* group | 0.398 | 0.004 |

**TABLE 3** Clinical characteristics of patients with periodontitis pre- and post-treatment[a,b]

| General characteristics | | Moderate periodontitis (n = 7) | | | | | Severe periodontitis (n = 7) | | | | |
|---|---|---|---|---|---|---|---|---|---|---|---|
| Age (years) | | 46.29 ± 5.33 | | | | | 51.71 ± 3.34 | | | | |
| Gender (male/female) | | 4/3 | | | | | 2/5 | | | | |
| | | Baseline | 1 M | 3 M | 6 M | P-value | Baseline | 1 M | 3 M | 6 M | P value |
| | PI | 1.42 ± 0.12 | 0.54 ± 0.08 | 0.50 ± 0.06 | 0.50 ± 0.11 | <0.001 | 0.97 ± 0.08 | 0.43 ± 0.06 | 0.40 ± 0.09 | 0.48 ± 0.11 | <0.001 |
| | PD | 2.77 ± 0.04 | 2.40 ± 0.05 | 2.46 ± 0.07 | 2.27 ± 0.08 | <0.001 | 3.45 ± 0.11 | 2.43 ± 0.13 | 2.41 ± 0.08 | 2.24 ± 0.06 | <0.001 |
| | CAL | 2.90 ± 0.06 | 2.58 ± 0.07 | 2.64 ± 0.04 | 2.44 ± 0.05 | <0.001 | 3.77 ± 0.23 | 2.95 ± 0.37 | 2.80 ± 0.20 | 2.72 ± 0.25 | <0.001 |
| | mSBI | 1.10 ± 0.15 | 0.53 ± 0.11 | 0.55 ± 0.09 | 0.51 ± 0.09 | <0.001 | 1.08 ± 0.06 | 0.29 ± 0.06 | 0.30 ± 0.03 | 0.27 ± 0.04 | <0.001 |
| Full mouth | GI | 2.11 ± 0.09 | 1.75 ± 0.09 | 1.81 ± 0.06 | 1.75 ± 0.07 | <0.001 | 2.16 ± 0.03 | 1.57 ± 0.06 | 1.61 ± 0.06 | 1.56 ± 0.06 | <0.001 |
| | BOP (%) | 71.80 ± 6.32 | 44.28 ± 7.85 | 48.48 ± 5.76 | 44.02 ± 5.65 | <0.001 | 72.41 ± 2.47 | 30.50 ± 5.56 | 29.57 ± 2.77 | 26.17 ± 3.46 | <0.001 |
| | Number of PDs ≥ 5 mm | 4.29 ± 0.87 | 0.57 ± 0.30 | 0.00 ± 0.00 | 0.00 ± 0.00 | <0.001 | 30.43 ± 6.57 | 6.71 ± 3.25 | 4.00 ± 1.02 | 2.00 ± 0.58 | <0.001 |
| | Total number of teeth | 27.43 ± 0.30 | 27.43 ± 0.30 | 27.43 ± 0.30 | 27.43 ± 0.30 | – | 26.86 ± 0.55 | 26.43 ± 0.72 | 25.71 ± 1.17 | 25.14 ± 1.28 | 0.183 |
| | PI | 1.64 ± 0.19 | 0.57 ± 0.24 | 0.79 ± 0.26 | 0.71 ± 0.27 | <0.001 | 1.00 ± 0.18 | 0.43 ± 0.14 | 0.21 ± 0.11 | 0.50 ± 0.17 | <0.001 |
| | PD | 3.21 ± 0.20 | 2.74 ± 0.15 | 2.69 ± 0.13 | 2.60 ± 0.13 | <0.001 | 3.29 ± 0.18 | 2.62 ± 0.13 | 2.60 ± 0.11 | 2.52 ± 0.14 | <0.001 |
| Tooth –subgingival plaque | CAL | 3.29 ± 0.19 | 2.79 ± 0.16 | 2.79 ± 0.17 | 2.79 ± 0.16 | <0.001 | 4.12 ± 0.18 | 3.76 ± 0.25 | 3.40 ± 0.24 | 3.81 ± 0.24 | <0.001 |
| | mSBI | 1.52 ± 0.19 | 0.62 ± 0.16 | 0.83 ± 0.15 | 0.74 ± 0.14 | <0.001 | 1.21 ± 0.18 | 0.40 ± 0.09 | 48 ± 0.08 | 0.43 ± 0.08 | <0.001 |
| | GI | 2.29 ± 0.18 | 1.93 ± 0.18 | 2.07 ± 0.10 | 2.00 ± 0.15 | 0.035 | 2.29 ± 0.18 | 1.64 ± 0.13 | 1.71 ± 0.13 | 1.64 ± 0.13 | 0.035 |

[a]Values are presented as mean ± standard deviation. The P value was obtained by Wilcoxon signed-rank test before and after periodontitis treatment. A P value of <0.05 was considered to indicate statistical significance.
[b]BOP, bleeding on probing; CAL, clinical attachment level; GI, gingival index; mSBI, modified sulcus bleeding index; PD, probing depth; PI, plaque index.

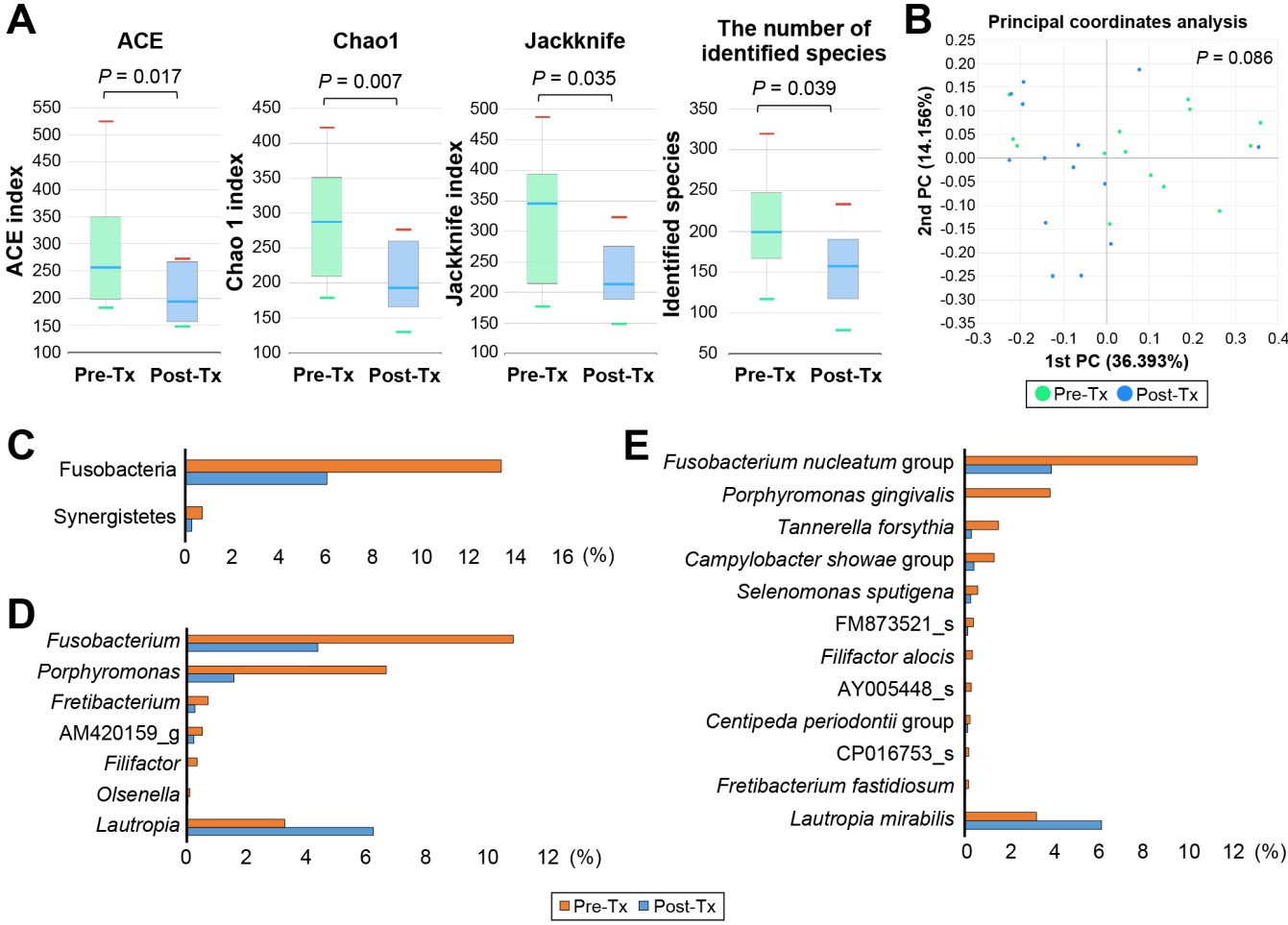

**FIG 3** Differences in subgingival microbiota before and after treatment of periodontitis. Subgingival samples were collected at baseline and 6 months following treatment from 14 subjects with periodontitis. (A) Comparison of ACE, Chao1, Jackknife, and the number of identified species before and after treatment. Each value is presented as a box plot. Top, middle, and bottom lines of the boxes represent the 25th, 50th (median), and 75th percentiles, respectively. The significance of differences pre- and post-treatment was evaluated using the Wilcoxon rank-sum test, and $P < 0.05$ was considered to indicate statistical significance. (B) PCoA plot illustrating beta diversity distance matrices of the Jensen–Shannon distance comparing the sample distribution pre- and post-treatment. Green dots represent pre-treatment samples, and blue dots represent post-treatment samples. Permutational multivariate analysis of variance results demonstrated beta set significance ($P = 0.086$) before vs after treatment. (C) Phyla showing significant differences pre- and post-treatment. (D) Genera showing significant differences pre- and post-treatment. The six genera listed at the top were dominant pre-treatment, while the genus listed at the bottom was dominant after treatment among genera >0.01% of all bacterial species. (E) Species showing statistically significant differences pre- and post-treatment. The 11 species listed at the top were dominant pre-treatment, while the species listed at the bottom was dominant after treatment among species >0.01% in the subgingival plaque samples. $P < 0.05$ by Kruskal–Wallis $H$ test comparing pre- and post-treatment. Pre-tx, pre-treatment; post-Tx, post-treatment.

with the extent of reduction in subgingival plaque; however, the sum of the reduction percentages of the three species showed a positive correlation between saliva and subgingival plaque samples ($r = 0.571$, $P = 0.033$) (Supplement 7).

## DISCUSSION

To determine whether subgingival microbial changes are reflected in the salivary microbiota, 16S rDNA amplicon sequencing of subgingival plaque and saliva samples was performed according to the severity of periodontal disease and before and after non-surgical periodontal treatment for periodontitis. There was a clear difference in the distribution of microbiota (beta diversity) between subgingival plaque and saliva samples, and saliva tended to have increased diversity (alpha diversity) compared to subgingival plaque. Nevertheless, most phyla in saliva and subgingival plaque showed a

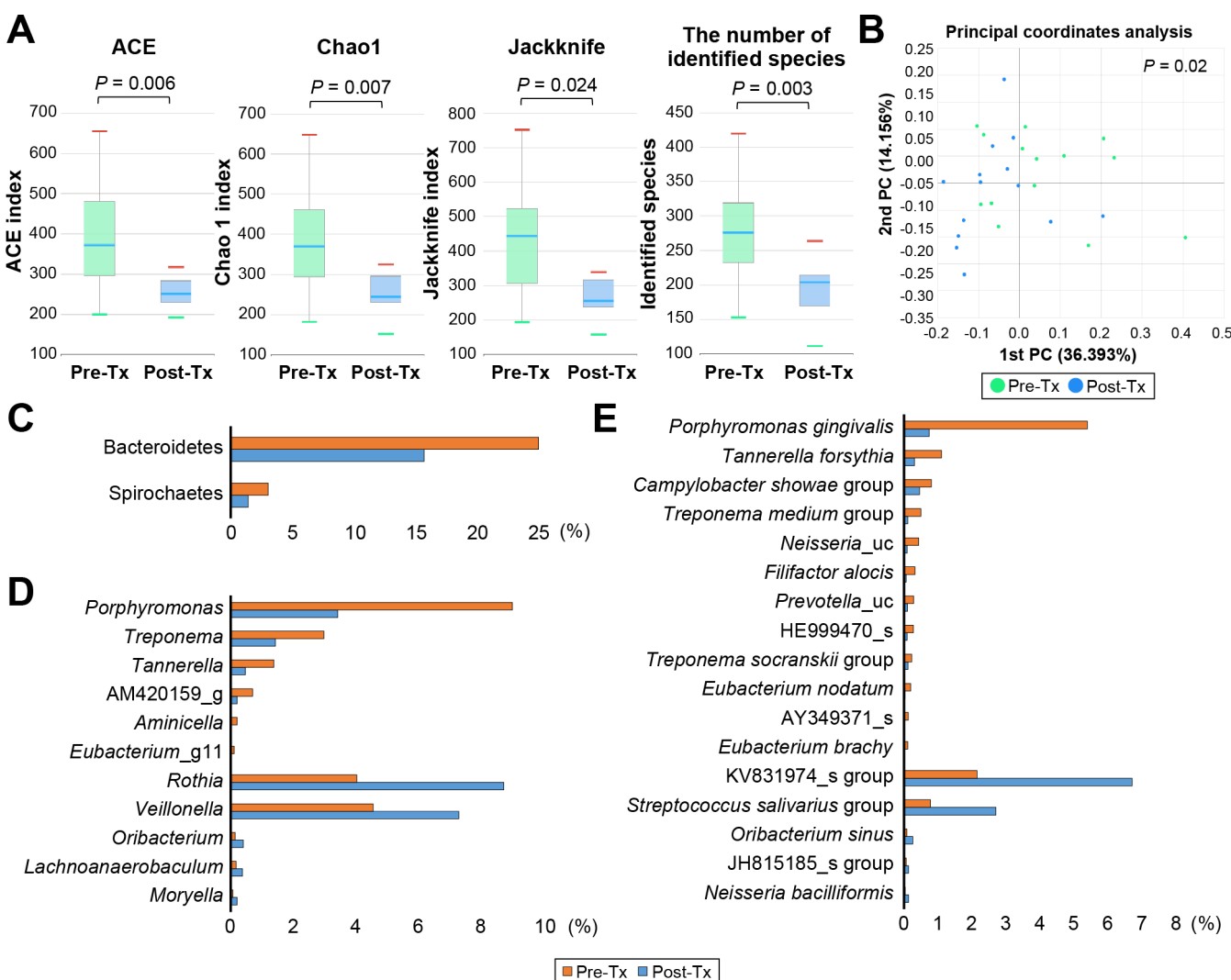

**FIG 4** Differences in salivary microbiota before and after treatment of periodontitis. Salivary samples were collected at baseline and 6 months following treatment of 14 subjects with periodontitis. (A) Comparison of ACE, Chao1, Jackknife, and the number of identified species pre-treatment and post-treatment. Each value is presented as a box plot. Top, middle, and bottom lines of the boxes represent the 25th, 50th (median), and 75th percentiles, respectively. The significance of differences before and after treatment was evaluated using the Wilcoxon rank-sum test, and $P < 0.05$ was considered to indicate a statistically significant difference. (B) PCoA plot illustrating beta diversity distance matrices of the Jensen–Shannon distance comparing the sample distribution pre- and post-treatment. Green dots represent pre-treatment samples, and the blue dots represent post-treatment samples. Permutational multivariate analysis of variance results demonstrated beta set significance ($P = 0.02$) before vs after treatment. (C) Phyla showing significant differences pre- and post-treatment (D) Genera showing significant differences pre- and post-treatment. The six genera listed at the top were dominant pre-treatment, while the five genera listed at the bottom were dominant after treatment among genera >0.01% of saliva samples. (E) Species showing significant differences pre- and post-treatment. The 11 species listed at the top were dominant pre-treatment, while the five species listed at the bottom were dominant after treatment among species >0.01% in the saliva samples. $P < 0.05$ by Kruskal–Wallis $H$ test comparing pre- and post-treatment samples. Pre-tx, pre-treatment; post-Tx, post-treatment.

positive correlation in distribution as disease severity increased, and the relative abundance of health- or periodontitis-dominant bacterial species in saliva was also correlated with that in subgingival plaque, consistent with the findings of previous reports (22–24). Kageyama et al. showed that the relative abundance of subgingival-specific taxa in saliva can reflect that in subgingival plaque, providing an indication of periodontal health (10). We found that non-surgical periodontal treatment, aimed at eliminating and subverting the subgingival microbiota, resulted in shifts toward the microbial signature of healthy controls, not only in subgingival plaque but also in saliva (Supplements 5 and 6). These results indicate that although the microbial composition

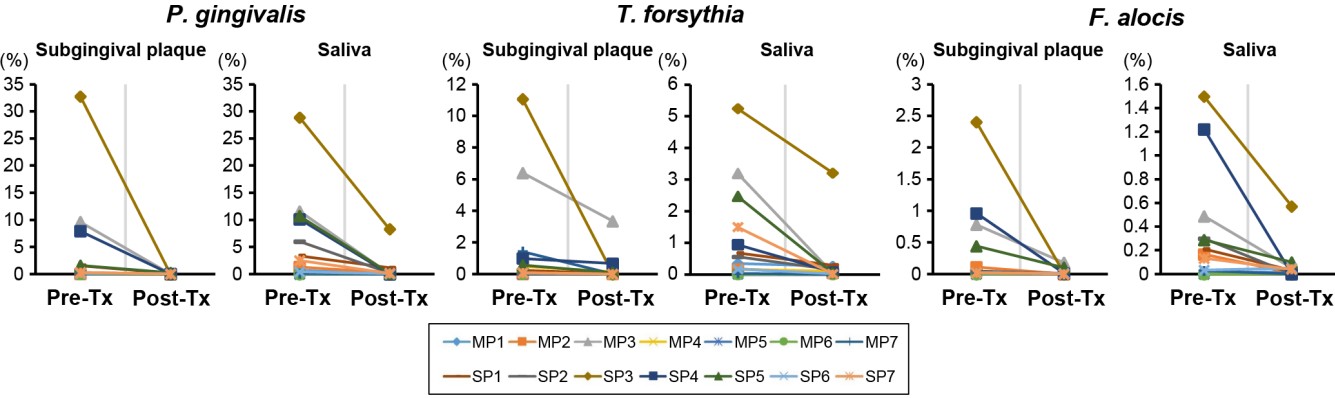

**FIG 5** Relative abundance of *P. gingivalis*, *T. forsythia*, and *F. alocis* pre- and post-treatment in subgingival plaque and saliva samples of seven moderate (MP1 ~7) and seven severe (SP1 ~7) periodontitis patients. The relative abundance of *P. gingivalis*, *T. forsythia*, and *F. alocis* in subgingival plaque and saliva samples was compared between pre- and post-treatment periodontitis samples. MP, moderate periodontitis patient; SP, severe periodontitis patient; pre-Tx, pre-treatment; post-Tx, post-treatment.

and diversity of subgingival plaque and saliva differ, the changes in characteristics of subgingival microbiota according to disease severity and following periodontal treatment are reflected in saliva.

As the severity of periodontitis increased, the inflammatory responses became more pronounced, creating a more favorable condition for inflammophilic pathogens that thrive on inflammatory products (2). Furthermore, the formation of deep periodontal pockets created an environment conducive to strict anaerobic bacteria, as these pockets provide low-oxygen conditions ideal for their growth (25). This distinct subgingival environment that changes with severity of periodontal disease is believed to drive the compositional differences observed between subgingival plaque and saliva (Fig. 1B). Moreover, clustering analysis by disease severity indicated that salivary and subgingival microbial communities tend to show greater compositional difference as the disease becomes more severe (Supplement 2). This finding is supported by the relative abundance of species comprising more than 1% of subgingival plaque (Fig. 2B) and saliva (Fig. 2C). Among species with an abundance of ≥1%, the number present simultaneously in both subgingival plaque and saliva was 6 in cases of severe periodontitis compared to 12 in healthy subjects. Despite the difference in composition, however, the relative abundance of periodontitis- or health-related species in subgingival plaque was correlated with that in saliva (Table 2; Supplement 4), although salivary levels tended to be lower than those in subgingival plaque (Supplement 3). This indicates that progression of periodontitis is accompanied by a quantitative increase of the total number of periodontitis-related species as well as a compositional shift in subgingival plaque (26), and this can result in a higher number of the subgingival bacteria spilling over into saliva. These findings support the use of periodontitis-related bacterial species in saliva as a diagnostic tool for periodontitis. However, because bacteria from other sites in the oral cavity mix with saliva during spillover from the subgingival plaque, the relative abundance of periodontitis-related species in saliva may be underestimated compared to their abundance in the subgingival plaque.

This study aimed to identify whether a subgingival microbial shift is reflected in saliva by analyzing subgingival and salivary microbiota following non-surgical periodontal treatment. To achieve this, we analyzed samples from subjects at time points expected to show significant differences in microbiota as a result of periodontitis treatment. The composition of subgingival plaque is significantly influenced by the depth of the periodontal pockets, with the abundance of periodontitis-related bacteria increasing with PD (25, 27). It is predictable that individuals with greater clinical improvement will also have greater improvements in their microbiota. Therefore, we preferentially selected subjects based on the degree of improvement in mean PD and the number

of sites with PD of ≥5 mm, and samples collected after 6 months of treatment were analyzed for post-treatment sequencing because the extent of clinical improvement was most evident at that time point (Table 3). Non-surgical periodontal treatment focused on subgingival plaque removal can induce ecological changes in the subgingival environment and cause alterations in the composition of the subgingival microbiota (5, 28, 29). As expected, periodontal treatment resulted in an immediate improvement in clinical parameters that was maintained for up to 6 months after treatment, and this was accompanied by a microbial shift in saliva as well as subgingival plaque. The alpha diversity of microbiota decreased and periodontitis-dominant taxa decreased, whereas health-dominant taxa increased following treatment in both subgingival plaque and saliva; the findings in subgingival plaque are partially consistent with those reported previously (30, 31). Our findings for saliva are consistent with a previous study that analyzed the salivary microbiota of four healthy controls and eight patients with severe periodontitis before and 3 months after treatment (15). The microbial shift following treatment is partially consistent with those reported by Belstrøm et al., who performed an interventional study with simultaneous characterization of subgingival and salivary microbiotas (32). They showed that periodontal treatment resulted in a significantly higher relative abundance of *Streptococcus*, *Rothia*, and *Actinomyces* spp. in combination with a significant decrease in *Porphyromonas* and *Treponema* spp. in subgingival plaque. However, they failed to show a microbial shift in saliva following periodontal treatment and concluded that periodontal treatment only had a minor impact on the major salivary microbiota. Moreover, the changes in subgingival plaque were gradually reversed until 12 weeks after treatment, and the significant impact of periodontal therapy on alpha diversity decreased after 2 and 6 weeks and was completely reversed after 12 weeks (32). The reason why shifts in the salivary microbiota were observed more clearly in the current study may be because we analyzed subjects with excellent clinical improvement as evaluated by mean PD reduction and sites with a PD of ≥5 mm 6 months after treatment. However, comparative analyses of the oral microbiota over time are needed for subjects who show significant clinical improvement vs those who do not improve.

Relative abundances of *P. gingivalis*, *T. forsythia*, and *F. alocis* in subgingival plaque were correlated with those in saliva, and levels of these bacteria in saliva as well as subgingival plaque were significantly reduced by non-surgical periodontal treatment. Moreover, the sum of the reduction ratios of the three bacterial species in saliva following treatment showed a positive correlation with that observed in subgingival plaque (Supplement 7). These results support our previous suggestion that these three bacterial species can function as salivary biomarkers for the severity of periodontal diseases in Koreans (14). However, the levels of reduction of periodontitis-dominant species by treatment were more remarkable in subgingival plaque than saliva samples (Fig. 5). We observed post-treatment levels comparable to that in the G group (Supplements 5 and 6). This is likely because clinical parameters after 6 months of treatment were similar to those in the G group (Table 1), making it reasonable to compare levels post-treatment with levels in gingivitis. We interpret these findings to indicate that periodontitis-dominant species are subgingival plaque dominant and that the ecological changes that make it difficult for bacteria to recolonize following treatment occur more strongly in the subgingival environment. Moreover, since these periodontitis bacteria can also be present in other spaces in the oral cavity, such as the tongue (33, 34), their reduction by non-surgical treatment may be masked in saliva. However, large-scale, long-term data are needed to determine what proportion of pathogenic bacteria are present in saliva and for how long the relative abundance of periodontitis-dominant species in saliva is maintained to be able to prevent disease recurrence.

The reduction ratios of each of the three bacterial species in saliva following treatment were not correlated with the reduction ratios of these species in subgingival plaque. This is presumed to be because the relative abundance levels following treatment in subgingival plaque and saliva were reduced to almost zero, regardless of levels before treatment (Fig. 5). Exceptionally, the relative abundances of these three

bacterial species were not decreased in subject SP3's saliva, although those in subgingival plaque decreased to levels similar to that seen in the other subjects (Fig. 5). To further investigate this finding, we examined the clinical characteristics of SP3. We noted improvements in the mean PD and the number of sites with a PD of ≥5 mm or more through extraction of five teeth that showed severe alveolar bone loss, but PI and parameters related to gingival inflammation remained relatively high, especially 3–6 months after treatment (Supplement 8). Although more data on how quickly pathogenic bacteria can recolonize the oral cavity depending on plaque control is needed, these results can be interpreted as rapid recolonization of the total oral cavity by pathogenic bacteria due to a lack of improvement in SP3's ability to control plaque after treatment. Saliva can act as a reservoir for recolonization into subgingival plaque (35–37), so if toothbrushing efficiency is not improved after treatment, subgingival plaque can quickly be recolonized by pathogenic bacteria, which may eventually lead to further loss of attachment.

We showed that subgingival microbial changes were reflected in salivary microbiota changes according to the severity of periodontal disease and following the non-surgical periodontal treatment of periodontitis. These results indicate that dysbiosis in the subgingival bacterial community during progression of periodontal disease and microbial shifts following treatment can affect the entire oral ecosystem. Thus, bacteria in saliva can reflect the level of periodontal health.

## ACKNOWLEDGMENTS

This work was supported by a grant from the National Research Foundation of Korea, funded by the Korea government (grant no. 2023R1A2C2004811) and a grant from the Korea Health Technology R&D Project through the Korea Health Industry Development Institute, funded by the Ministry of Health & Welfare, Republic of Korea (grant no. HI23C0409).

## AUTHOR AFFILIATIONS

[1]Department of Periodontology, Institute of Oral Health Science, Ajou University School of Medicine, Suwon, South Korea
[2]Department of Oral Biochemistry, Korean Collection for Oral Microbiology, School of Dentistry, Chosun University, Gwangju, South Korea

## AUTHOR ORCIDs

Jae-Suk Jung http://orcid.org/0000-0002-2018-6450
Suk Ji http://orcid.org/0000-0001-9720-6731

## FUNDING

| Funder | Grant(s) | Author(s) |
| --- | --- | --- |
| National Research Foundation of Korea (NRF) | 2023R1A2C2004811 | Suk Ji |
| Korea Health Industry Development Institute (KHIDI) | HI23C0409 | Suk Ji |

## AUTHOR CONTRIBUTIONS

Jae-Suk Jung, Conceptualization, Data curation, Formal analysis, Investigation, Methodology, Project administration, Supervision, Validation, Visualization, Writing – original draft, Writing – review and editing | Joong-Ki Kook, Conceptualization, Data curation, Formal analysis, Investigation, Methodology, Project administration, Resources, Software, Supervision, Validation, Visualization, Writing – original draft, Writing – review and editing | Soon-Nang Park, Conceptualization, Data curation, Methodology, Resources, Software | Yun Kyong Lim, Data curation, Formal analysis, Methodology, Resources, Software, Validation, Visualization | Geum Hee Choi, Data curation, Investigation,

Methodology, Project administration, Resources, Software, Supervision, Validation, Visualization | Sunjin Kim, Data curation, Formal analysis, Methodology, Software | Suk Ji, Conceptualization, Data curation, Formal analysis, Funding acquisition, Investigation, Methodology, Project administration, Resources, Software, Supervision, Validation, Visualization, Writing – original draft, Writing – review and editing

## ETHICS APPROVAL

This study was approved by the Institutional Review Board for Human Subjects of Ajou University Dental Hospital (AJOUIRB-SMP-2018–062). All participants were recruited from Ajou University Dental Hospital from May 2018 to March 2020, and informed consent was obtained from all subjects.

## ADDITIONAL FILES

The following material is available online.

### Supplemental Material

**Supplement 1 (Spectrum01030-24-s0001.pdf).** Comparison of ACE, Chao1, Jackknife, and the number of identified species.
**Supplement 2 (Spectrum01030-24-s0002.pdf).** B.PCoA plot illustrating beta diversity distance matrices of the Jensen-Shannon distance.
**Supplement 3 (Spectrum01030-24-s0003.pdf).** Relative abundance of the 20 species.
**Supplement 4 (Spectrum01030-24-s0004.pdf).** A heat map distribution of the 20 species.
**Supplement 5 (Spectrum01030-24-s0005.pdf).** Pre- and post-treatment levels of taxa showing significant difference for all 14 periodontitis patients in subgingival plaque.
**Supplement 6 (Spectrum01030-24-s0006.pdf).** Pre- and post-treatment levels of taxa showing saliva samples from 14 periodontitis patients.
**Supplement 7 (Spectrum01030-24-s0007.pdf).** Correlation in rate of reduction of phyla, genera, and species in subgingival plaque and saliva samples.
**Supplement 8 (Spectrum01030-24-s0008.pdf).** Panoramic radiographs of patient.

### Open Peer Review

**PEER REVIEW HISTORY (review-history.pdf).** An accounting of the reviewer comments and feedback.

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
