## [Reviewer comments · Microbiology Spectrum]

Microbiology Spectrum

Salivary microbiota reflecting changes in subgingival microbiota

Suk Ji, Jae-Suk Jung, Joong-Ki Kook, Soon-Nang Park, Yun Kyong Lim, Geum Hee Choi, and Sunjin Kim

Corresponding Author(s): Suk Ji, Ajou University Medical Center

Review Timeline:

Submission Date:	April 24, 2024
Editorial Decision:	June 1, 2024
Revision Received:	July 18, 2024
Accepted:	September 3, 2024

Editor: Deena Altman

Reviewer(s): The reviewers have opted to remain anonymous.

Transaction Report:

DOI: <https://doi.org/10.1128/spectrum.01030-24>

Re: Spectrum01030-24 (Salivary microbiota reflecting changes in subgingival microbiota)

Dear Prof. Suk Ji:

Thank you for the privilege of reviewing your work. Below you will find my comments, instructions from the Spectrum editorial office, and the reviewer comments.

Revision Guidelines

Sincerely,
Deena Altman
Editor
Microbiology Spectrum

Reviewer #1 (Comments for the Author):

I reviewed the manuscript entitled "Salivary microbiota reflecting changes in subgingival microbiota" by Jung et al. In this study, Jung et al. analyzed and compared the subgingival and salivary microbiota from human subjects divided into 4 groups: healthy, gingivitis, moderate periodontitis and aggressive periodontitis. The study is of interest since saliva samples are easier to collect, are non-invasive and allows higher compliance from subject donors. The manuscript is well written and well designed. My comments are described below:

1) Lines 57-58: Periodontitis is undoubtedly induced by subgingival biofilm and the presence of keystone pathogens, such as *Porphyromonas gingivalis*. However, periodontal tissue damage is also caused by exacerbated host immune responses, such as neutrophils and macrophages recruitment, secretion of metalloproteinases and pro-inflammatory molecules. Even though the focus of this manuscript is the subgingival microbiota, the authors should still revise this sentence and include host cell responses as one of the causes for periodontitis. Below are some studies on this topic:

- <https://www.ncbi.nlm.nih.gov/pmc/articles/PMC4276050/>
- <https://pubmed.ncbi.nlm.nih.gov/35029093/>
- <https://www.ncbi.nlm.nih.gov/pmc/articles/PMC4510669/>

2) Lines 102-130: In the methods section, were all patients examined by the same clinician to avoid subjective data collection? If the samples were collected by different clinicians, were the clinicians calibrated to perform the exams and sample collection? This piece of information needs to be added for this referee and in the the methods section.

3) Line 152: Please add reference.

4) Fig 1A, Fig 2A, Fig 3A, Fig 4A, Fig 5: Can the authors add the unit used for the y-axis in all these graphs?

5) Fig1B: Does this image illustrate that the bacteria found in saliva were different than the bacteria found in subgingival plaques? The authors should clarify this question to this referee and in the description of the results, and add more information about Fig1B result interpretation to the reader.

6) Lines 214-219: Please add the results/graphs that the authors are referring to in this sentence.

7) Lines 234-235: Fig 1B shows that the bacterial distribution is essentially different between saliva and subgingival plaque samples. Can the authors affirm that "the microbial distribution of subgingival plaque is correlated with that of saliva" based on their results in Figs 1 and 2? Would it be more accurate to describe that there is a correlation of microbial distribution between saliva and subgingival plaque when examining only specific key pathogens in the healthy and severe periodontitis group? Please clarify and add the appropriate revision and explanation to the manuscript.

8) Lines 239-241: Why only a few subjects were selected for follow up analyses? Why not all patients in each group?

9) Lines 264-267: "(...) were significantly decreased following treatment in (...)". Please revise this sentence. The word "in" might need to be removed.

Reviewer #2 (Comments for the Author):

Spectrum01030-24-authors-review

The authors present a valuable attempt to investigate whether changes in the salivary microbiota reflect the severity of periodontal disease and the effects of non-surgical periodontal treatment. While the authors concluded that the salivary microbiota could mirror changes in the subgingival microbiota, some data suggest otherwise.

However, some aspects require improvement:

In the supplementary data, include a table detailing the percentage relative abundance of species described in the correlation analysis, such as Table 2. It would be interesting to see the positive correlations and percentage range differences between plaque and saliva for those species.

Given that periodontitis is a biofilm-associated disease, the manuscript should discuss how saliva might provide a representation but potentially underestimate the actual numbers found in subgingival plaque.

These additions would enhance the manuscript by clarifying the differences between plaque and salivary microbiota, thereby supporting the conclusions drawn.

In the subsection titled "Microbial characteristics of subgingival and saliva samples before and after periodontitis treatment," it should be clearly stated that the aim is to determine the changes in microbial distribution in both saliva and plaque samples before treatment and six months after non-surgical periodontal treatment.

For the flow of the text author may rewrite the text in a different order. It would be better if in line 248 instead "Relative abundance..." they add the text in line 253, "Alpha-diversity.... saliva samples (Fig. 4B, PERMANOVA results, $p = 0.02$)" Then add the "Relative abundances... (Supplement 4 and 5)." Then introduce the "Levels of Fusobacteria...."

Although the authors mention that clinical parameters were measured and saliva and subgingival plaque samples were collected at 1-, 3-, and 6- months post-treatment, the results only focus on the last time point. Did the authors analyze the data from the other time points? It would be interesting to compare what happens early after treatment versus at six months. Did the clinical

parameters decline from 1 month to 6 months? Were the microbial shifts already present at one month and three months, and how do they compare to the 6-month data?

Including this information, or at least discussing it, would significantly enhance the manuscript. If the authors only focused on the 6-month data, it should be explained why. The materials and methods section suggests a longitudinal analysis, but the discussion does not address it further. Clarifying this aspect would provide a more comprehensive understanding of the study's findings.

In Figure 1B, it would be beneficial to highlight the samples based on their disease status. This would allow us to observe how the healthy, gingivitis, moderate disease, and severe periodontitis samples cluster within the total dataset.

Supplementary Figure 2 clearly demonstrates that saliva and plaque communities cluster separately with increasing disease severity. While these communities are quite mixed in healthy individuals, a distinct separation becomes evident as pathology worsens. How do the authors discuss this finding in the context of using saliva as a sample source for severe periodontitis, given that it may not accurately reflect the microbial composition of plaque?

This observation is further supported by Figure 3B and 3C, where the species distribution in healthy individuals and those with gingivitis is similar. However, in cases of moderate and severe periodontitis, the microbial communities in plaque and saliva differ significantly.

Spectrum01030-24-authors-review

The authors present a valuable attempt to investigate whether changes in the salivary microbiota reflect the severity of periodontal disease and the effects of non-surgical periodontal treatment. While the authors concluded that the salivary microbiota could mirror changes in the subgingival microbiota, some data suggest otherwise.

However, some aspects require improvement:

In the supplementary data, include a table detailing the percentage relative abundance of species described in the correlation analysis, such as Table 2. It would be interesting to see the positive correlations and percentage range differences between plaque and saliva for those species.

Given that periodontitis is a biofilm-associated disease, the manuscript should discuss how saliva might provide a representation but potentially underestimate the actual numbers found in subgingival plaque.

These additions would enhance the manuscript by clarifying the differences between plaque and salivary microbiota, thereby supporting the conclusions drawn.

In the subsection titled "***Microbial characteristics of subgingival and saliva samples before and after periodontitis treatment,***" it should be clearly stated that the aim is to determine the changes in microbial distribution in both saliva and plaque samples before treatment and six months after non-surgical periodontal treatment.

For the flow of the text author may rewrite the text in a different order. It would be better if in line 248 instead "*Relative abundance...*" they add the text from in line 253, "*Alpha-diversity.... saliva samples (Fig. 4B, PERMANOVA results, $p = 0.02$)*"

Then add the "*Relative abundances... (Supplement 4 and 5).*" Then introduce the "*Levels of Fusobacteria....*"

Although the authors mention that clinical parameters were measured and saliva and subgingival plaque samples were collected at 1-, 3-, and 6- months post-treatment, the results only focus on the last time point. Did the authors analyze the data from the other time points? It would be interesting to compare what happens early after treatment versus at six months. Did the clinical parameters decline from 1 month to 6 months? Were the microbial shifts already present at one month and three months, and how do they compare to the 6-month data?

Including this information, or at least discussing it, would significantly enhance the manuscript. If the authors only focused on the 6-month data, it should be explained why. The materials and methods section suggests a longitudinal analysis, but the discussion does not address it further. Clarifying this aspect would provide a more comprehensive understanding of the study's findings.

In Figure 1B, it would be beneficial to highlight the samples based on their disease status. This would allow us to observe how the healthy, gingivitis, moderate disease, and severe periodontitis samples cluster within the total dataset.

Supplementary Figure 2 clearly demonstrates that saliva and plaque communities cluster separately with increasing disease severity. While these communities are quite mixed in healthy individuals, a distinct separation becomes evident as pathology worsens. How do the authors discuss this finding in the context of using saliva as a sample source for severe periodontitis, given that it may not accurately reflect the microbial composition of plaque?

This observation is further supported by Figure 3B and 3C, where the species distribution in healthy individuals and those with gingivitis is similar. However, in cases of moderate and severe periodontitis, the microbial communities in plaque and saliva differ significantly.

Additionally, in Figures 3C, 3D, and 3E, as well as Figures 4C, 4D, and 4E, the phyla present in plaque and saliva samples are different, and the most affected species also vary. For instance, Fusobacteria, particularly *Fusobacterium nucleatum*, dominate in plaque, whereas *Porphyromonas gingivalis*, along with other genera like *Rothia* and *Veillonella*, are more prevalent in saliva.

Although there is a reduction in periodontitis-associated species in both plaque and saliva, the specific species present in each environment differ slightly. This suggests that saliva may not accurately represent the microbial composition of plaque in severe pathological states. The authors need to further discuss these differences and their implications for using saliva as a diagnostic tool in severe periodontitis.

The authors raise the question of how quickly pathogenic bacteria can recolonize the oral cavity. Although they collected samples at early time points (1- and 3-months post-treatment), they neither present nor discuss this data in the manuscript. That would be a valuable addition to the manuscript.

Salivary microbiota reflecting changes in subgingival microbiota

Jae-Suk Jung, Joong-Ki Kook, Soon-Nang Park, Yun Kyong Lim, Geum Hee Choi, Sunjin Kim, Suk Ji

Response to Reviewer #1: I reviewed the manuscript entitled "Salivary microbiota reflecting changes in subgingival microbiota" by Jung et al. In this study, Jung et al. analyzed and compared the subgingival and salivary microbiota from human subjects divided into 4 groups: healthy, gingivitis, moderate periodontitis and aggressive periodontitis. The study is of interest since saliva samples are easier to collect, are non-invasive and allows higher compliance from subject donors. The manuscript is well written and well designed.

My comments are described below:

→ We thank Reviewer #1 for your time and expert opinion.

Comment 1 Lines 57-58: Periodontitis is undoubtedly induced by subgingival biofilm and the presence of keystone pathogens, such as *Porphyromonas gingivalis*. However, periodontal tissue damage is also caused by exacerbated host immune responses, such as neutrophils and macrophages recruitment, secretion of metalloproteinases and pro-inflammatory molecules. Even though the focus of this manuscript is the subgingival microbiota, the authors should still revise this sentence and include host cell responses as one of the causes for periodontitis. Below are some studies on this topic:

- <https://www.ncbi.nlm.nih.gov/pmc/articles/PMC4276050/>
- <https://pubmed.ncbi.nlm.nih.gov/35029093/>
- <https://www.ncbi.nlm.nih.gov/pmc/articles/PMC4510669/>

Response: Thank for mentioning the part about “the host response” that we missed. The sentence was improved on Page 4 Lines 55-57; **Periodontal disease is an inflammatory disease that affects the supporting structures of the teeth and is**

characterized by the host's immune response to microorganisms in the subgingival space (1-3).

Comment 2 Lines 102-130: In the methods section, were all patients examined by the same clinician to avoid subjective data collection? If the samples were collected by different clinicians, were the clinicians calibrated to perform the exams and sample collection? This piece of information needs to be added for this referee and in the methods section.

Response: All participants received a full-mouth periodontal examination by a single trained periodontist, and saliva and subgingival sampling was also performed by the same periodontist.

The information has been modified as follows (Page 6, Line 111 ~113); **All participants received a full-mouth periodontal examination followed by saliva and subgingival plaque sampling by a single trained periodontist.**

Comment 3 Line 152: Please add reference.

Response: The reference has been added (Page 8, Line 151 ~ 152); **DNA extraction and sequencing were performed at ChunLab, Inc ((<https://www.cjbioscience.com/>)) as described in our previous study (14).**

Comment 4 Fig 1A, Fig 2A, Fig 3A, Fig 4A, Fig 5: Can the authors add the unit used for the y-axis in all these graphs?

Response: All figures that you pointed out have been revised.

+

Comment 5 Fig1B: Does this image illustrate that the bacteria found in saliva were different than the bacteria found in subgingival plaques? The authors should clarify this question to this referee and in the description of the results, and add more information about Fig1B result interpretation to the reader.

Response: Regarding Fig 1B and Supplement 2, Reviewer #2 also mentioned, so the content was modified as follows (Page 10, Line 204 ~ Page 11, line 212); **Beta diversity of clustering analysis based on Jensen-Shannon divergence at the species level, analyzed across 51 subjects and by disease severity, showed different communities in the subgingival plaque and saliva samples (PERMANOVA results, $p = 0.001$) (Fig. 1B). Clustering analysis by severity of periodontal disease showed that salivary and subgingival communities tended to cluster separately as severity increased. In healthy individuals, the subgingival and salivary microbial communities were mixed ($p=0.043$), but a clear tendency toward separation emerged as severity worsened ($p < 0.001$ of SP group, Supplement 2).**

Comment 6 Lines 214-219: Please add the results/graphs that the authors are referring to in this sentence.

Response: The figure description corresponding to the sentence has been supplemented as follow (Page 11, Line 216 ~ 219); As the severity of periodontal disease increased, Firmicutes, Proteobacteria, and Actinobacteria tended to decrease, while Bacteroidetes, Fusobacteria, Spirochaetes, and Synergistetes tended to increase in both subgingival plaque and saliva samples (Fig. 2A).

Comment 7 Lines 234-235: Fig 1B shows that the bacterial distribution is essentially different between saliva and subgingival plaque samples. Can the authors affirm that "the microbial distribution of subgingival plaque is correlated with that of saliva" based on their results in Figs 1 and 2? Would it be more accurate to describe that there is a correlation of microbial distribution between saliva and subgingival plaque when examining only specific key pathogens in the healthy and severe periodontitis group? Please clarify and add the appropriate revision and explanation to the manuscript.

Response: The sentence was modified to clarify the meaning and improved as follows (Page 12, Line 235 ~ 236); **These results indicate that the distribution of periodontitis- or health-dominant species in subgingival plaque is correlated with that of saliva.**

Comment 8 Lines 239-241: Why only a few subjects were selected for follow up analyses? Why not all patients in each group?

Response: The purpose of this study was to determine whether subgingival microbial changes following the non-surgical periodontal treatment of periodontitis are reflected in the saliva microbiota. To achieve this, we thought it would be best to analyze subjects who are expected to experience significant changes in their microbiota as a result of periodontitis treatment. It is predictable that individuals with greater clinical improvement will also have greater improvements in their microbiota. Therefore, we first analyzed the degree of improvement in clinical parameters after 1, 3, and 6 months, and preferentially selected subjects for post-treatment sequencing based on the degree of improvement in mean probing depth (PD) and number of sites with $PD \geq 5$

mm.

To explain the reasons for the subject selection, the text was improved as follows (Page 16, Line 331 ~ Page 17, Line 341); **This study aimed to identify whether a subgingival microbial shift is reflected in saliva by analyzing subgingival and salivary microbiota following non-surgical periodontal treatment. To achieve this, we analyzed samples from subjects at time points expected to show significant differences in microbiota as a result of periodontitis treatment. The composition of subgingival plaque is significantly influenced by the depth of the periodontal pockets, with the abundance of periodontitis-related bacteria increasing with PD (25, 27). It is predictable that individuals with greater clinical improvement will also have greater improvements in their microbiota. Therefore, we preferentially selected subjects based on the degree of improvement in mean PD and number of sites with PD \geq 5 mm, and samples collected after 6 months of treatment were analyzed for post-treatment sequencing because the extent of clinical improvement was most evident at that time point (Table 3).**

Comment Lines 264-267: "(...) were significantly decreased following treatment in (...)".
9 Please revise this sentence. The word "in" might need to be removed.

Response: The word "in" was removed as following (Page 13, Line 266 ~ 270 **Levels of six genera in subgingival plaque, including *Fusobacterium* and *Porphyromonas*, were significantly decreased following treatment, while levels of six genera including *Treponema* and *Porphyromonas* were significantly**

decreased in saliva samples following treatment (Fig. 3D and 4D, respectively).

Response to Reviewer #2: The authors present a valuable attempt to investigate whether changes in the salivary microbiota reflect the severity of periodontal disease and the effects of non-surgical periodontal treatment. While the authors concluded that the salivary microbiota could mirror changes in the subgingival microbiota, some data suggest otherwise.

However, some aspects require improvement:

→ We thank Reviewer #2 for your time and expert opinion.

Comment 1 In the supplementary data, include a table detailing the percentage relative abundance of species described in the correlation analysis, such as Table 2. It would be interesting to see the positive correlations and percentage range differences between plaque and saliva for those species.

Response: Relative abundances of the periodontitis- and health-dominant species by the severity of periodontal disease shown in Table 2 were added as supplement 3.

The sentence was also improved as following (Page 11, Line 228 ~230);

Examination of the bacterial species present at greater than 1% in any of the four groups revealed 13 species increased (periodontitis-dominant) whereas seven species decreased (health-dominant) as the severity of periodontal disease increased (Supplement 3).

Supplement 3. Relative abundance of the 20 species present at levels greater than 1% of the bacteria that differed among the four groups by disease severity.

Relative abundance according to disease severity in subgingival plaque (Sub-P) and saliva samples was expressed as mean \pm standard deviation.

Comment 2 Given that periodontitis is a biofilm-associated disease, the manuscript should discuss how saliva might provide a representation but potentially underestimate the actual numbers found in subgingival plaque. These additions would enhance the manuscript by clarifying the differences between plaque and salivary microbiota, thereby supporting the conclusions drawn.

Response: Beta diversity between communities of saliva and subgingival plaque according to the severity of disease depth showed significant differences, which means that the bacterial composition of both environments is different. Nevertheless, periodontitis- or health-related bacterial species showed a positive correlation in the distribution of saliva and subgingival plaque samples. These results can be interpreted as periodontitis- or health-related bacterial species in saliva can be spill-over from subgingival plaque and so the dysbiosis process occurring in the subgingival environment can also be reflected in saliva.

The discussion was enhanced as follows (Page 16, Line 320 ~330);
Despite the difference in composition, however, the relative abundance of periodontitis- or health-related species in subgingival plaque was correlated with that in saliva (Table 2 and Supplement 4), although salivary levels tended to be lower than those in subgingival plaque (Supplement 3). This indicates that progression of periodontitis is accompanied by a quantitative increase of the total number of periodontitis-related species as well as a compositional shift in subgingival plaque (26), and this can result in a higher number of the subgingival bacteria spilling over into saliva. These findings support the use of periodontitis-related bacterial species in saliva as a diagnostic tool for periodontitis. However, because bacteria from other sites in the oral cavity mix with saliva during spillover from the subgingival plaque, the relative

abundance of periodontitis-related species in saliva may be underestimated compared to their abundance in the subgingival plaque.

Comment 3 In the subsection titled "Microbial characteristics of subgingival and saliva samples before and after periodontitis treatment," it should be clearly stated that the aim is to determine the changes in microbial distribution in both saliva and plaque samples before treatment and six months after non-surgical periodontal treatment.

Response: The sentence has been improved as following (Page 12, Line 251 ~ Line 253); To determine whether changes in microbial distribution were observed not only in subgingival plaque, which is the target site of non-surgical periodontal treatment, but also in saliva, microbial shifts pre- and post-treatment were compared for each of the subgingival plaque and saliva samples. Because the extent of clinical improvement was most evident after 6 months of treatment (Table 3), subgingival plaque (Figure 3) and saliva (Figure 4) samples at that time point were analyzed using 16S rDNA amplicon sequencing.

Comment 4 For the flow of the text author may rewrite the text in a different order. It would be better if in line 248 instead "Relative abundance..." they add the text from in line 253, "Alpha-diversity.... saliva samples (Fig. 4B, PERMANOVA results, $p = 0.02$)" Then add the "Relative abundances... (Supplement 4 and 5)." Then introduce the "Levels of Fusobacteria...."

Response: As per your suggestion, the order of the sentences has been changed (Page 13, Line 256 ~ 263); Alpha diversity was significantly higher pre-treatment

compared with post-treatment in both subgingival plaque and saliva samples (Fig. 3A and 4A). Clustering analysis at the species level showed a significant difference between pre- and post-treatment saliva samples (Fig. 4B, PERMANOVA results, $p = 0.02$). **Relative abundances of the taxa that differed significantly pre- and post-treatment for all 14 subjects are presented in Figures 3C-E and Figures 4C-E.** Additionally, to assess the extent of changes after treatment in moderate and severe periodontitis groups, the relative abundances of the taxa are displayed along with those in the H and G groups (Supplement 5 and 6).

Comment 5 Although the authors mention that clinical parameters were measured and saliva and subgingival plaque samples were collected at 1-, 3-, and 6- months post-treatment, the results only focus on the last time point. Did the authors analyze the data from the other time points? It would be interesting to compare what happens early after treatment versus at six months. Did the clinical parameters decline from 1 month to 6 months? Were the microbial shifts already present at one month and three months, and how do they compare to the 6-month data?

Including this information, or at least discussing it, would significantly enhance the manuscript. If the authors only focused on the 6-month data, it should be explained why. The materials and methods section suggests a longitudinal analysis, but the discussion does not address it further. Clarifying this aspect would provide a more comprehensive understanding of the study's findings.

Response: Clinical parameters were measured at 1-, 3-, and 6- months post-treatment and clinical improvement in all parameters (PI, PD, CAL, mSBI, GI, BOP(%), number of PDs ≥ 5 mm) was observed following nonsurgical treatment

(Table 3). Samplings of saliva and subgingival plaque were collected at 1-, 3-, and 6- months post-treatment, however the sequencing analysis was performed for 6-month samples.

Relevant content has been presented in the discussion section (Page 16, Line 331 ~ Page 17, Line 341); **This study aimed to identify whether a subgingival microbial shift is reflected in saliva by analyzing subgingival and salivary microbiota following non-surgical periodontal treatment. To achieve this, we analyzed samples from subjects at time points expected to show significant differences in microbiota as a result of periodontitis treatment. The composition of subgingival plaque is significantly influenced by the depth of the periodontal pockets, with the abundance of periodontitis-related bacteria increasing with PD (25, 27). It is predictable that individuals with greater clinical improvement will also have greater improvements in their microbiota. Therefore, we preferentially selected subjects based on the degree of improvement in mean PD and number of sites with $PD \geq 5$ mm, and samples collected after 6 months of treatment were analyzed for post-treatment sequencing because the extent of clinical improvement was most evident at that time point (Table 3).**

Comment In Figure 1B, it would be beneficial to highlight the samples based on their disease status. This would allow us to observe how the healthy, gingivitis, moderate disease, and severe periodontitis samples cluster within the total dataset.

Response: Figure 1B has been revised based on your comment. It shows how the distribution of microbiota in subgingival plaque and saliva of subjects with

health, gingivitis, moderate periodontitis, and severe periodontitis cluster within the total dataset.

Comment 7 Supplementary Figure 2 clearly demonstrates that saliva and plaque communities cluster separately with increasing disease severity. While these communities are quite mixed in healthy individuals, a distinct separation becomes evident as pathology worsens. How do the authors discuss this finding in the context of using saliva as a sample source for severe periodontitis, given that it may not accurately reflect the microbial composition of plaque? This observation is further supported by Figure 2B and 2C, where the species distribution in healthy individuals and those with gingivitis is similar. However, in cases of moderate and severe periodontitis, the microbial communities in plaque and saliva differ significantly.

Response: Thank you for your insightful comments. Relevant contents have been presented in Discussion section (Page 15, Line 308 ~ Page 16, Line 320); **As the severity of periodontitis increased, the inflammatory responses became more pronounced, creating a more favorable condition for inflammophilic pathogens that thrive on inflammatory products (2). Furthermore, the formation of deep periodontal pockets created an environment conducive to strict anaerobic bacteria, as these pockets provide low-oxygen conditions ideal for their growth (25). This distinct subgingival environment that changes with severity of periodontal disease is believed to drive the compositional differences observed between subgingival plaque and saliva (Fig. 1B). Moreover, clustering analysis by disease severity indicated that salivary and subgingival microbial communities tend to show greater compositional difference as the disease becomes more severe (Supplement 2). This finding is supported by the relative abundance of species comprising more than 1% of subgingival plaque (Fig. 2B) and saliva (Fig. 2C); among species with abundance $\geq 1\%$, the number present simultaneously in both subgingival plaque and saliva was 6 in cases of severe periodontitis compared to 12 in healthy subjects.**

Re: Spectrum01030-24R1 (Salivary microbiota reflecting changes in subgingival microbiota)

Dear Prof. Suk Ji:

Your manuscript has been accepted, and I am forwarding it to the ASM production staff for publication. Your paper will first be checked to make sure all elements meet the technical requirements. ASM staff will contact you if anything needs to be revised before copyediting and production can begin. Otherwise, you will be notified when your proofs are ready to be viewed.

Please add dedicated section titled "Data Availability" at the end of Materials and Methods section

Sincerely,
Deena Altman
Editor
Microbiology Spectrum

Reviewer #1 (Public repository details (Required)):

The authors provided information on where to find the gene sequencing.

Reviewer #1 (Comments for the Author):

All my comments have been addressed. I have no further comments.

Reviewer #2 (Public repository details (Required)):

The study has focused on 16S rRNA sequencing of saliva and dental plaque. Depending on the journal guidelines, the authors may be required to share their sequencing raw data in a public repository or database.

Reviewer #2 (Comments for the Author):

Dear authors,

The revised manuscript submitted by the authors successfully addresses the reviewer's comments. The improvements have strengthened the study, which presents a valuable exploration of how salivary microbiota reflects the severity of periodontal disease and the effects of non-surgical periodontal treatment.

I have no further comments at this time.